# Genetics of height and risk of atrial fibrillation: A Mendelian randomization study

**Michael G. Levin**[1,2,3], **Renae Judy**[4], **Dipender Gill**[5,6,7,8,9], **Marijana Vujkovic**[2,3], **Shefali S. Verma**[10,11], **Yuki Bradford**[10,11], **Regeneron Genetics Center**[12¶], **Marylyn D. Ritchie**[10,11], **Matthew C. Hyman**[1,2], **Saman Nazarian**[1,2], **Daniel J. Rader**[2,10,13], **Benjamin F. Voight**[10,13,14], **Scott M. Damrauer**[3,4]*

**1** Division of Cardiovascular Medicine, University of Pennsylvania Perelman School of Medicine, Philadelphia, Pennsylvania, United States of America, **2** Department of Medicine, University of Pennsylvania Perelman School of Medicine, Philadelphia, Pennsylvania, United States of America, **3** Corporal Michael J. Crescenz VA Medical Center, Philadelphia, Pennsylvania, United States of America, **4** Department of Surgery, University of Pennsylvania Perelman School of Medicine, Philadelphia, Pennsylvania, United States of America, **5** Department of Epidemiology and Biostatistics, School of Public Health, Imperial College London, London, United Kingdom, **6** Centre for Pharmacology & Therapeutics, Department of Medicine, Imperial College London, London, United Kingdom, **7** Novo Nordisk Research Centre Oxford, Oxford, United Kingdom, **8** Clinical Pharmacology and Therapeutics Section, Institute of Medical and Biomedical Education and Institute for Infection and Immunity, St George's, University of London, London, United Kingdom, **9** Clinical Pharmacology Group, Pharmacy and Medicines Directorate, St George's University Hospitals NHS Foundation Trust, London, United Kingdom, **10** Department of Genetics, University of Pennsylvania Perelman School of Medicine, Philadelphia, Pennsylvania, United States of America, **11** Institute for Biomedical Informatics, University of Pennsylvania Perelman School of Medicine, Philadelphia, Pennsylvania, United States of America, **12** Tarrytown, New York, United States of America, **13** Institute for Translational Medicine and Therapeutics, University of Pennsylvania Perelman School of Medicine, Philadelphia, Pennsylvania, United States of America, **14** Department of Systems Pharmacology and Translational Therapeutics, University of Pennsylvania Perelman School of Medicine, Philadelphia, Pennsylvania, United States of America

¶ Membership of the Regeneron Genetics Center is provided in the Acknowledgments.
* damrauer@upenn.edu

**Data Availability Statement:** GWAS summary statistics for height are publicly available, and can be downloaded from the GIANT consortium

## Abstract

### Background

Observational studies have identified height as a strong risk factor for atrial fibrillation, but this finding may be limited by residual confounding. We aimed to examine genetic variation in height within the Mendelian randomization (MR) framework to determine whether height has a causal effect on risk of atrial fibrillation.

### Methods and findings

In summary-level analyses, MR was performed using summary statistics from genome-wide association studies of height (GIANT/UK Biobank; 693,529 individuals) and atrial fibrillation (AFGen; 65,446 cases and 522,744 controls), finding that each 1-SD increase in genetically predicted height increased the odds of atrial fibrillation (odds ratio [OR] 1.34; 95% CI 1.29 to 1.40; $p = 5 \times 10^{-42}$). This result remained consistent in sensitivity analyses with MR methods that make different assumptions about the presence of pleiotropy, and when accounting for the effects of traditional cardiovascular risk factors on atrial fibrillation. Individual-level phenome-wide association studies of height and a height genetic risk score

website (https://portals.broadinstitute.org/collaboration/giant/index.php/GIANT_consortium_data_files). Summary statistics for atrial fibrillation were contributed by the AFGen consortium (http://afgen.org), are publicly available, and may be downloaded from the Variant to Function Knowledge Portal (http://www.kp4cd.org/datasets/v2f). Individual-level data from the Penn Medicine BioBank are not publicly available due to their sensitive nature, however the data may be made available with the appropriate ethical approval and data sharing agreements (biobank@upenn.edu).

**Funding:** BFV was supported by the National Institute of Diabetes and Digestive and Kidney Diseases (R01-DK101478; https://www.niddk.nih.gov/) and a Linda Pechenik Montague Investigator Award (upenn.edu). SMD was supported by the US Department of Veterans Affairs (IK2-CX001780; VA.gov). This publication does not represent the views of the Department of Veterans Affairs or the United States government. DG was supported by funding from the Wellcome Trust (https://wellcome.ac.uk/). Genetic sequencing of Penn Medicine Biobank participants was performed in collaboration with Regeneron Genetics Center (https://www.regeneron.com/genetics-center), who also reviewed the manuscript but had no role in study design, data analysis, or decision to publish. The other funders had no role in study design, data collection and analysis, decision to publish, or preparation of the manuscript.

**Competing interests:** The authors have declared that no competing interests exist.

**Abbreviations:** GRS, genetic risk score; GWAS, genome-wide association study; HDL, high-density lipoprotein; LDL, low-density lipoprotein; MR, Mendelian randomization; OR, odds ratio; PheWAS, phenome-wide association study.

were performed among 6,567 European-ancestry participants of the Penn Medicine Biobank (median age at enrollment 63 years, interquartile range 55–72; 38% female; recruitment 2008–2015), confirming prior observational associations between height and atrial fibrillation. Individual-level MR confirmed that each 1-SD increase in height increased the odds of atrial fibrillation, including adjustment for clinical and echocardiographic confounders (OR 1.89; 95% CI 1.50 to 2.40; $p = 0.007$). The main limitations of this study include potential bias from pleiotropic effects of genetic variants, and lack of generalizability of individual-level findings to non-European populations.

## Conclusions

In this study, we observed evidence that height is likely a positive causal risk factor for atrial fibrillation. Further study is needed to determine whether risk prediction tools including height or anthropometric risk factors can be used to improve screening and primary prevention of atrial fibrillation, and whether biological pathways involved in height may offer new targets for treatment of atrial fibrillation.

## Author summary

### Why was this study done?

- Studies have identified height as a risk factor for atrial fibrillation, a common abnormal heart rhythm.

- Whether being taller actually elevates risk of atrial fibrillation, or if this association is an artifact of prior study designs, remains unclear.

### What did the researchers do and find?

- We examined randomly allocated genetic variants associated with height within the Mendelian randomization framework to study the effects of height on risk of atrial fibrillation.

- Genetic variants associated with taller stature were also associated with increased risk of atrial fibrillation. This finding was consistent across multiple analysis methods, including when accounting for other known atrial fibrillation risk factors.

### What do these findings mean?

- Taller individuals are likely to be at increased risk of atrial fibrillation.

- Future research is needed to better define the pathways connecting height to atrial fibrillation.

## Introduction

Atrial fibrillation is a common cardiac arrhythmia, with a population prevalence of 0.5%, affecting more than 33 million individuals worldwide [1]. A number of risk factors are associated with atrial fibrillation, including chronic diseases like chronic kidney disease, heart failure, thyroid disease, obesity, obstructive sleep apnea, sleep apnea, and valvular heart disease, as well as cardiac surgery, smoking, and anthropometric factors [2–5]. Recent studies have identified both common and rare genetic variants at more than 100 independent loci contributing to the incidence of atrial fibrillation, and heritability is estimated at 20% [6–8]. Even with treatment, affected individuals are at risk of cardioembolic stroke, heart failure, and death [2].

Height has been identified as a risk factor for a number of cardiometabolic diseases, including coronary artery disease, atrial fibrillation, and venous thromboembolism [9,10]. The relationship between height and atrial fibrillation in particular has been identified in large observational studies, with greater height strongly associated with increased risk of atrial fibrillation [4,11–18]. These studies are limited in assessing the causality and potential mediators or confounders of this association by their observational design, and randomized controlled trials of anthropometric traits are not feasible. Although a number of factors influence height including childhood illness/development, diet/nutrition, and socioeconomic status, height has a strong genetic component. Large genome-wide association studies (GWASs) have provided heritability estimates of 60%–70%, and have identified more than 700 independent loci that contribute to height [19–22].

In the current study, we utilize human genetic data within the Mendelian randomization (MR) framework to evaluate a potential causal association between height and atrial fibrillation. MR analysis exploits the random assortment of genetic variants during meiosis as an instrumental variable to estimate the causal relationship between a trait and an outcome of interest. Here, we use summary data from large, multiethnic GWASs of height (693,529 individuals) and atrial fibrillation (65,446 cases and 522,744 controls) to estimate the effect of genetically predicted height on risk of atrial fibrillation [7,19]. We then use participant-level data from the Penn Medicine Biobank within the observational phenome-wide association study (PheWAS) framework to identify other clinical associations with height, and within the single-sample MR framework to further assess the impact of height on atrial fibrillation after adjustment for clinical risk factors.

## Methods

This study is reported per the Strengthening the Reporting of Observational Studies in Epidemiology (STROBE) guideline (S1 STROBE Checklist). The study did not have a pre-registered or published analysis plan. All analyses were planned prior to study initiation.

### Two-sample MR analysis

Summary-level data for height was obtained from a 2018 meta-analysis of GWASs of height [19]. This analysis was a fixed-effects meta-analysis combining results from a GWAS of height performed among 456,426 participants from the UK Biobank (adjusted for age, sex, recruitment center, genotyping batch, and 10 genetic principal components) and results from a 2014 GWAS published by the GIANT (Genetic Investigation of ANthropometric Traits) Consortium, which included 253,288 participants from 79 studies (adjusted for age, sex, and study-specific covariates). Genetic variants associated with height at genome-wide significance ($p < 5 \times 10^{-8}$) were then LD-pruned (distance threshold = 10,000 kb, $r^2 = 0.001$) using the *clump_data* command in the *TwoSampleMR* package in R to identify an independent set of variants to serve as a genetic instrument for height [23]. The independent variants associated

with height at genome-wide significance ($p < 5 \times 10^{-8}$) were then harmonized with variants from the 2018 Roselli et al. [7] atrial fibrillation GWAS from the AFGen (Atrial Fibrillation Genetics) Consortium, using the default settings of the *harmonize_data* command in the *TwoSampleMR* package in R to ensure that effect estimates were aligned to the same allele. This study included 65,446 atrial fibrillation cases and 522,744 controls from more than 50 studies (84.2% European, 12.5% Japanese, 2% African American, and 1.3% Brazilian and Hispanic), including participants from UK Biobank, Biobank Japan, other international biobanks, and international cardiovascular cohort studies (adjusted for age, sex, and study-specific covariates) [7]. In total, 707 independent SNPs associated with height were available in the atrial fibrillation outcome GWAS. These 707 SNPs accounted for 11.2% of the variability in height (S1 Table).

Inverse-variance-weighted 2-sample MR was used as the primary analysis, with weighted median, MR-Egger, and MR-PRESSO performed as sensitivity analyses to account for potential violations of the instrumental variable assumptions [24]. Further sensitivity analysis was performed using a genetic instrument for height constructed excluding any SNPs nominally ($p < 0.05$) associated with coronary artery disease, high-density lipoprotein (HDL), low-density lipoprotein (LDL), total cholesterol, triglycerides, fasting glucose, fasting insulin, diabetes, BMI, waist-to-hip ratio, and systolic blood pressure, all identified using the MR-Base database [25–28].

For each variant included in the genetic instruments, variance ($R^2$) was calculated using the formula $R^2 = 2 \times \text{MAF} \times (1 - \text{MAF}) \times \text{beta}^2$ (where MAF represents the effect allele frequency and beta represents the effect estimate of the genetic variant in the exposure GWAS). *F*-statistics were then calculated for each variant using the formula $F = \frac{R^2 \times (N-2)}{1 - R^2}$ (where $R^2$ represents the variance in exposure explained by the genetic variant, and $N$ represents the number of individuals in the exposure GWAS) to assess the strength of the selected instruments [29]. Bias due to sample overlap between the exposure and outcome GWASs was estimated and found to be negligible (S1 Methods) [30]. The estimates for the effect of height on atrial fibrillation are reported per 1–standard deviation (SD) increase in height. A scaled estimate per 10-cm increase in height has also been calculated for the main analysis based on the February 2020 UK Biobank population standard deviation of height (9.28 cm).

To account for the possibility that the restrictive genetic instrument may introduce collider or ascertainment bias by conditioning on associated traits, multivariable MR was performed using the *TwoSampleMR* package in R. This method allows the direct effects of multiple traits on an outcome to be determined jointly. The effect of height on atrial fibrillation was estimated in analyses that individually adjusted for potential confounders, including coronary artery disease, HDL, LDL, total cholesterol, triglycerides, fasting glucose, fasting insulin, diabetes, BMI, waist-to-hip ratio, and systolic blood pressure, using genetic variants obtained from MR-Base as above. Potential confounders that had significant ($p < 0.05$) direct effects on atrial fibrillation in the individual multivariable models were then combined in a final model to jointly estimate their direct effects on atrial fibrillation.

Bidirectional MR was performed to assess for the possibility of reverse causality. A genetic instrument for atrial fibrillation containing 73 independent, genome-wide significant variants was constructed using the same methods as for the height instrument. Inverse-variance-weighted, weighted median, and MR-Egger analyses were performed.

## Height genetic risk score

A standardized, weighted genetic risk score (GRS) for height was calculated for each individual in the Penn Medicine Biobank using imputed dosage information in an additive model,

weighted by the effect size of each independent (distance threshold = 10,000 kb, $r^2$ = 0.001), genome-wide significant ($p < 5 \times 10^{-8}$) variant from the 2018 Yengo et al. GWAS of height [19], using the *clump_data* command in the *TwoSampleMR* package in R. Scores were standardized, centered to mean = 0, and scaled.

## PheWAS

PheWASs were performed to identify clinical diagnoses associated both measured height and the height GRS, using the default setting of the *PheWAS* package in R [31]. Briefly, international classification disease codes obtained from the electronic health record were mapped to "Phecodes," and individuals were assigned case/control status or excluded using default mapping parameters. Association between height/height GRS and phenotypes was tested using logistic regression, adjusted for age, $age^2$, sex, and 10 genetic principal components. The 10 genetic principal components included in the PheWAS analysis were selected a priori, without investigation of the variation of the data. For the more detailed individual-level MR analyses, 6 genetic principal components were identified qualitatively using a scree plot as explaining the majority of variation in the data (S4 Fig). The Bonferroni method was used to account for multiple testing ($p$ = 0.05/1,816 phenotypes).

## Individual-level MR

To estimate the population-averaged effect of increased height on risk of atrial fibrillation, individual-level MR was performed using the 2-stage method [32]. This analysis included 6,567 individuals with available height, genotype, and clinical covariate information. The instrumental variable was a GRS for height, computed from independent, genome-wide significant variants, weighted by effect on height in the 2018 GIANT/UK Biobank GWAS meta-analysis. Of the 707 genetic variants included in the main instrumental variable for height in the 2-sample MR analysis, 695 were available for use in the individual-level analysis (S3 Table). In the first stage of the 2-stage process, a linear regression was fitted with height as the dependent variable, and the GRS for height as the independent variable, among the subset of individuals without atrial fibrillation, adjusted for age, sex, and 6 genetic principal components. The 6 genetic principal components were selected qualitatively using a scree plot as explaining the majority of variation in the data (S4 Fig). In the second stage, a logistic regression model with robust standard errors was fit for both atrial fibrillation cases and controls, incorporating the fitted values of height from the first stage, adjusted for age, sex, and the 6 genetic principal components, with atrial fibrillation as the outcome. In additional models, each stage was adjusted for clinical diagnoses of hypertension, coronary artery disease, heart failure, hyperlipidemia, diabetes, chronic kidney disease, stroke, valvular heart disease, sleep apnea, thyroid disease, cardiac surgery, smoking, and echocardiographic left atrial size to account for potential cofounders/mediators of the height–atrial fibrillation relationship. These risk factors were selected for being common cardiometabolic risk factors or having prior associations with atrial fibrillation [2]. Sensitivity analysis was performed including only individuals who had available echocardiographic data.

## MR assumptions

MR relies on the naturally random assortment of genetic variants at conception to provide unbiased estimates of the effect of an exposure (in this case height) on an outcome (atrial fibrillation). To provide unbiased estimates, MR depends on 3 key assumptions [33]. First, genetic instruments must be associated with the exposure of interest (relevance assumption). Weak instruments that are not strongly associated with the exposure of interest may result in biased

estimates. To estimate the strength of the height genetic instrument, the *F*-statistic was calculated (S1 Table). The second assumption is that no confounders of the SNP–outcome association are present. This assumption is not readily testable because some confounders may not be measured, although at both the 2-sample and individual-level the MR analyses were adjusted for common cardiovascular risk factors. Further, because genetic variants are randomly allocated at conception, confounders of the SNP–outcome association should be distributed across the population. The third assumption is that the effect of the genetic instrument on atrial fibrillation is entirely through the effect on height (exclusion restriction). We performed a PheWAS using the GRS for height to assess for potentially pleiotropic effects of height. Further, we applied multiple MR methods including weighted median, MR-Egger, and MR-PRESSO, which are more robust to the presence of horizontal pleiotropy, a potential violation of the exclusion restriction assumption [24]. Finally, MR assumes that genetically mediated effects are similar to exogenous effects. In this case, height is highly heritable, although the effect of changes in stature related to environmental exposures (e.g., fetal growth restriction, malnutrition) may differ from the genetic effects assessed here.

## Penn Medicine Biobank

The Penn Medicine Biobank is a longitudinal genomics and precision medicine study in which participants consent to linkage of genomic information and biospecimens to the electronic health record. More than 60,000 individuals are currently enrolled. This study included 6,567 individuals of European ancestry (genetically determined) who underwent genotyping and had available electronic health record data. For individual-level analyses in the Penn Medicine Biobank, continuous demographic variables were summarized by mean and standard deviation, with categorical variables summarized by count and group percent. The Wilcoxon rank sum test and Fisher's exact test, respectively, were used to assess for significant differences between groups.

## Phenotype ascertainment

For individual-level analyses in Penn Medicine Biobank, phenotypes were determined by querying the electronic health record. International Classification of Diseases (ICD)–9/10 and Current Procedural Terminology (CPT) codes, in addition to laboratory measurements and vital signs, were used to identify height, weight, BMI, smoking status, diagnoses of heart failure, hypertension, diabetes mellitus, chronic kidney disease, sleep apnea, stroke, thyroid disease, valvular heart disease, and cardiac surgery. We identified 6,567 participants with available high-quality genotype and electronic health record phenotype data, with transthoracic echocardiogram data available for a subset of 2,842 individuals. Atrial fibrillation was defined using ICD-9/10 codes 427.31, I48.0, I48.1, I48.2, and I48.91. Atrial fibrillation ascertainment in the 2018 Roselli et al. GWAS meta-analysis was study-specific [7]. Data were extracted as of January 2017 (S1 Methods).

## Ethical approval

The investigators from the Penn Medicine Biobank (MGL, RJ, MV, SSV, YB, MDR, DJR, BFV, SMD) have received approval from the University of Pennsylvania Institutional Review Board. No additional approval was required for analyses of publicly available summary statistics.

## Statistical analysis

All statistical analyses were performed using R version 3.5.1 [34]. For statistical analyses, $p < 0.05$ was considered statistically significant.

## Results

### Population-level MR

To characterize the relationship between increasing height and risk of atrial fibrillation, we constructed a genetic instrument for height using 707 independent SNPs associated with height at a genome-wide level of significance ($p < 5 \times 10^{-8}$), which accounted for 11.2% of the variability in height (S1 Table). The mean $F$-statistic was 110 (range 22–948), suggesting the risk of weak instrument bias was low [30]. We performed 2-sample MR using summary statistics from a GWAS of atrial fibrillation including 65,446 atrial fibrillation cases and 522,744 controls. Inverse-variance-weighted modeling identified a significant association between increasing height and atrial fibrillation (odds ratio [OR] 1.34 per 1-SD increase in height; 95% CI 1.29 to 1.40; $p = 5 \times 10^{-42}$) (Fig 1). This corresponds to an OR of 1.37 (95% CI 1.31 to 1.44) per 10-cm increase in height among UK Biobank participants. The intercept from Egger

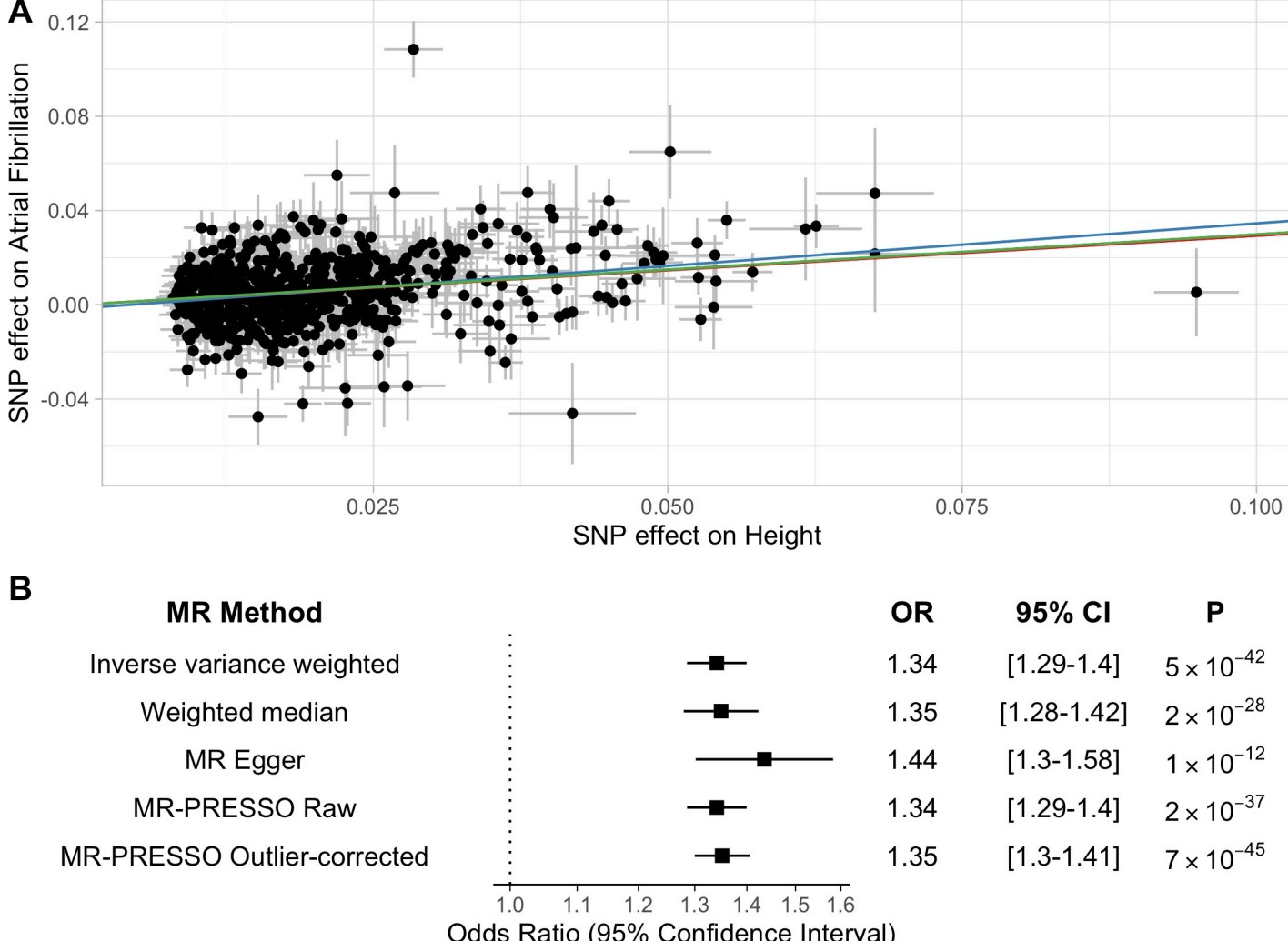

**Fig 1. Two-sample Mendelian randomization (MR).** Two-sample MR was performed using a genetic instrument containing 707 independent SNPs associated with height. (A) Each point represents the SNP effects on height and atrial fibrillation. Colored lines represent inverse-variance-weighted (red), weighted median (green), and MR-Egger (blue) estimates of the association between a 1-SD increase in height and risk of atrial fibrillation. (B) Odds ratios (ORs), 95% confidence intervals (CIs), and $p$-values for MR estimates.

regression was −0.001 ($p$ = 0.13), thus not providing evidence for significant pleiotropic bias. Similar estimates were obtained in sensitivity analyses from weighted median, MR-Egger, and MR-PRESSO models.

An additional genetic instrument for height was constructed, excluding variants nominally associated ($p < 0.05$) with potentially pleiotropic risk factors for atrial fibrillation. The selection of SNPs using a nominal $p$-value association of 0.05 was an arbitrarily chosen threshold to more liberally account for potentially pleiotropic effects of height-associated SNPs. The restrictive instrument consisted of 224 independent SNPs, in sum explaining 2.8% of the variance in height (S2 Table). The mean $F$-statistic was 88 (range 22–867). MR results using this restrictive genetic instrument were similar (S1 Fig).

We next performed multivariable MR. Height remained significantly associated with atrial fibrillation after adjustment for the effect of genetic variants separately on each of coronary artery disease, HDL, LDL, total cholesterol, triglycerides, fasting glucose, fasting insulin, diabetes, BMI, waist-to-hip ratio, and systolic blood pressure (S4 Table). This analysis identified significant associations of BMI, systolic blood pressure, total cholesterol, and coronary artery disease with atrial fibrillation after adjustment for height. The effect of height on risk of atrial fibrillation was similar in models accounting for these risk factors individually, and in a combined model jointly considering genetic variants associated with height, BMI, systolic blood pressure, total cholesterol, and coronary artery disease (Fig 2 and S5 Table).

To assess for the possibility of reverse causation, where genetic variants primarily associated with increased risk of atrial fibrillation may represent invalid instruments for height, a bidirectional MR analysis was performed. Using a genetic instrument including 73 independent, genome-wide significant variants associated with atrial fibrillation, there was no evidence for a causal association with height (S2 Fig).

## Demographics

Individual-level analysis focused on 6,567 European-ancestry individuals from the Penn Medicine Biobank with available genotype data linked to the electronic health record, recruited between 2008 and 2015. Individuals with atrial fibrillation were older (mean 66 versus 60 years; $p < 0.001$) than individuals without atrial fibrillation. Individuals with atrial fibrillation were taller (mean 174 versus 170 cm; $p < 0.001$), and significantly more likely to have been diagnosed with chronic kidney disease, coronary artery disease, heart failure, hyperlipidemia, hypertension, prior cardiac surgery, sleep apnea, smoking, stroke, or valvular heart disease, compared to individuals without atrial fibrillation (Table 1 and S6–S8). Among the subset of

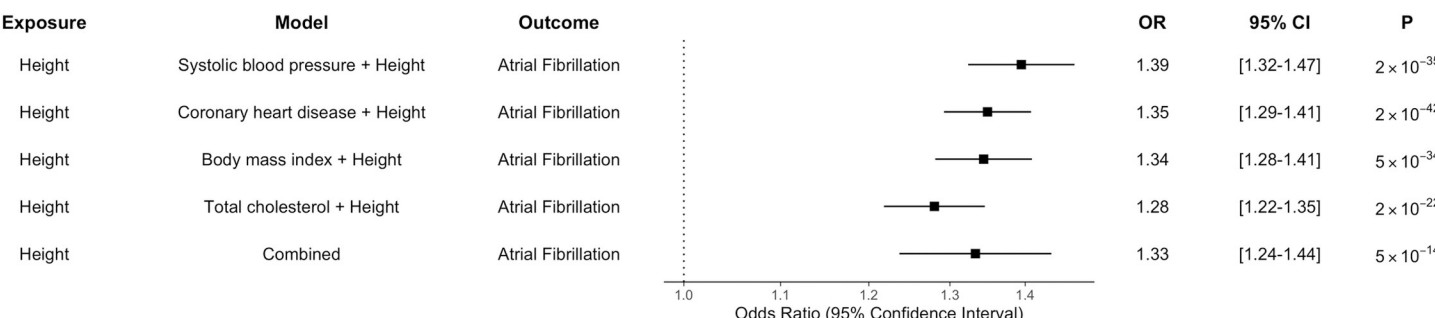

| Exposure | Model | Outcome | OR | 95% CI | P |
|---|---|---|---|---|---|
| Height | Systolic blood pressure + Height | Atrial Fibrillation | 1.39 | [1.32-1.47] | $2 \times 10^{-35}$ |
| Height | Coronary heart disease + Height | Atrial Fibrillation | 1.35 | [1.29-1.41] | $2 \times 10^{-42}$ |
| Height | Body mass index + Height | Atrial Fibrillation | 1.34 | [1.28-1.41] | $5 \times 10^{-34}$ |
| Height | Total cholesterol + Height | Atrial Fibrillation | 1.28 | [1.22-1.35] | $2 \times 10^{-22}$ |
| Height | Combined | Atrial Fibrillation | 1.33 | [1.24-1.44] | $5 \times 10^{-14}$ |

**Fig 2. Multivariable Mendelian randomization analysis.** Multivariable Mendelian randomization was performed to jointly consider the effect of genetic variants for cardiometabolic traits and height on atrial fibrillation. The effect of a 1-SD increase in height on risk of atrial fibrillation estimated by each model is displayed. ORs per 1-SD increase in height, 95% confidence intervals (CIs), and $p$-values for Mendelian randomization estimates are displayed.

**Table 1. Demographics.**

| Characteristic | Statistic | No atrial fibrillation N = 3,538 | Atrial fibrillation N = 3,029 | p-Value |
|---|---|---|---|---|
| Age (years) | Mean (SD) | 60 (14) | 66 (12) | <0.001 |
| Sex (female) | n (%) | 1,577 (45%) | 930 (31%) | <0.001 |
| BMI (kg/m²) | Mean (SD) | 29 (6) | 29 (6) | <0.001 |
| Height (cm) | Mean (SD) | 170 (11) | 174 (11) | <0.001 |
| Weight (kg) | Mean (SD) | 83 (21) | 88 (22) | <0.001 |
| Hypertension | n (%) | 1,799 (51%) | 2,117 (70%) | <0.001 |
| Coronary artery disease | n (%) | 1,747 (49%) | 1,764 (58%) | <0.001 |
| Heart failure | n (%) | 971 (27%) | 1,690 (56%) | <0.001 |
| Hyperlipidemia | n (%) | 1,829 (52%) | 1,946 (64%) | <0.001 |
| Diabetes | n (%) | 596 (17%) | 497 (16%) | 0.7 |
| Chronic kidney disease | n (%) | 400 (11%) | 577 (19%) | <0.001 |
| Sleep apnea | n (%) | 411 (12%) | 634 (21%) | <0.001 |
| Stroke | n (%) | 575 (16%) | 724 (24%) | <0.001 |
| Thyroid disease | n (%) | 71 (2.0%) | 78 (2.6%) | 0.14 |
| Cardiac surgery | n (%) | 830 (23%) | 2,063 (68%) | <0.001 |
| Valve disease | n (%) | 902 (25%) | 1,407 (46%) | <0.001 |
| Smoking | n (%) | 1,654 (47%) | 1,526 (50%) | 0.004 |
| Left atrial diameter (cm)* | Mean (SD) | 3.92 (0.76) | 4.39 (0.81) | <0.001 |

Demographics of individuals in Penn Medicine Biobank cohort with and without atrial fibrillation.

*Left atrial diameter available for 2,842 participants.

individuals with echocardiograms, individuals with atrial fibrillation had larger left atrial diameter (mean 4.39 versus 3.92 cm; $p < 0.001$).

## PheWASs of height and height GRS

To determine the association between genetically predicted height and clinical diagnoses, we first constructed a GRS for height using weights derived from the 2018 GIANT/UK Biobank height GWAS meta-analysis. PheWASs were performed to identify clinical diagnoses associated with both measured height and the height GRS across the Penn Medicine Biobank. Each 1-SD increase in measured height was associated with increased risk of atrial fibrillation (OR 1.55; 95% CI 1.41 to 1.71; $p = 3.6 \times 10^{-18}$) and decreased risk of coronary atherosclerosis (OR 0.66; $p = 6.9 \times 10^{-18}$). Each 1-SD increase in height GRS was associated with increased risk of atrial fibrillation and flutter (OR 1.16; 95% CI 1.09 to 1.23; $p = 7 \times 10^{-6}$) (Fig 3).

## Individual-level MR

Individual-level MR was performed in Penn Medicine Biobank participants to further assess the association between height and atrial fibrillation. Using the height GRS as an instrumental variable, height was significantly associated with atrial fibrillation (OR 1.75 per 1-SD increase in height; 95% CI 1.53 to 2.0; $p = 6 \times 10^{-5}$) after adjusting for age, sex, and 6 genetic principal components (Fig 4). This corresponds to an OR of 1.66 (95% CI 1.46 to 1.86) per 10-cm increase in height among Penn Medicine Biobank participants. Results were similar in sex-stratified analysis. Height remained associated with atrial fibrillation after adjustment for weight, hypertension, coronary artery disease, heart failure, hyperlipidemia, diabetes, chronic kidney disease, sleep apnea, stroke, thyroid disease, smoking, cardiac surgery, and valvular

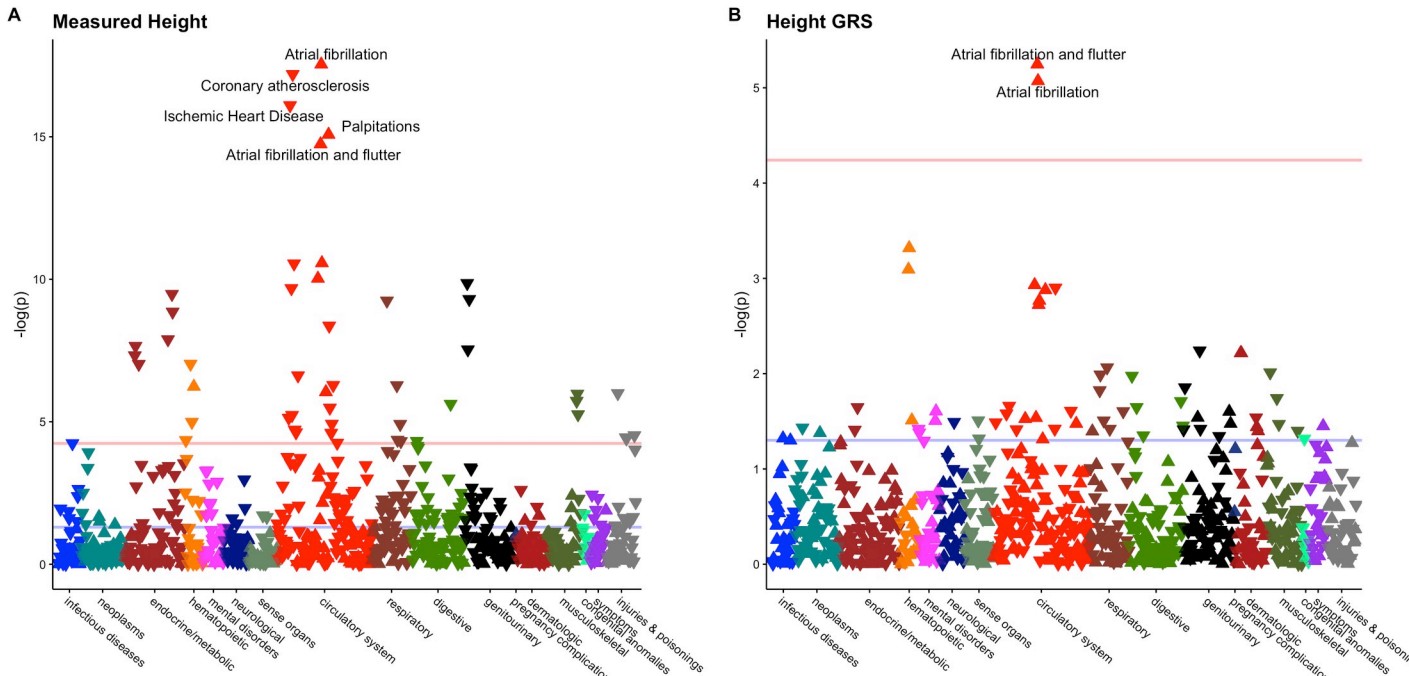

**Fig 3. Phenome-wide associations of clinical diagnoses with height and height genetic risk score (GRS).** Phenome-wide association studies were performed in Penn Medicine Biobank participants to identify clinical phenotypes associated with increased (A) height and (B) height GRS. The horizontal red line denotes the Bonferroni-adjusted level of significance (0.05/1,816 phenotypes), the blue line denotes the nominal level of significance (0.05), and triangles denote the direction of association between increasing height and risk of the phenotype (pointing upward = increased risk; pointing downward = decreased risk).

heart disease. After further adjustment for left atrial size, height remained significantly associated with atrial fibrillation (OR 1.83; 95% CI 1.44 to 2.31; $p$ = 0.01). When individual-level MR was restricted to the subset of individuals with complete data, the effect estimates were similar, with modest attenuation with sequential adjustment for clinical risk factors and left atrial size (S3 Fig).

## Discussion

In this study we used both population- and individual-level genetic information to test the association between height and atrial fibrillation. At the population level, there was a strong causal association between genetic determinants of height and risk of atrial fibrillation. This finding was robust to multiple sensitivity analyses of the MR methods and the genetic

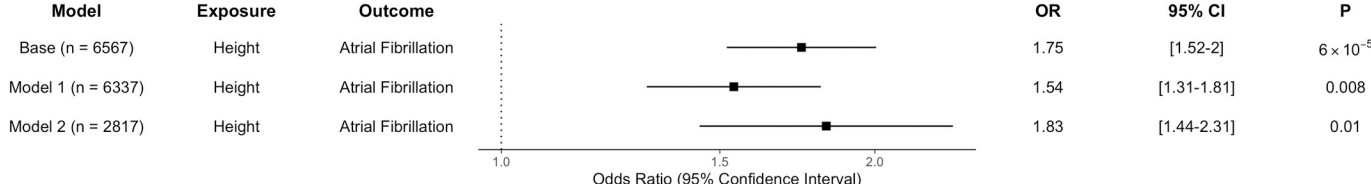

**Fig 4. Individual-level instrumental variable analysis.** Individual-level instrumental variable analysis was performed in Penn Medicine Biobank participants, using a weighted genetic risk score for height as an instrumental variable for measured height. The base model was adjusted for age, sex, and 6 genetic principal components. Model 1 was additionally adjusted for weight, hypertension, coronary artery disease, heart failure, hyperlipidemia, diabetes, chronic kidney disease, sleep apnea, stroke, thyroid disease, smoking, cardiac surgery, and valvular heart disease. Model 2 was additionally adjusted for left atrial size as measured on transthoracic echocardiogram. Odds ratios (ORs) are reported per 1-SD increase in height.

instrument for height. Observational analysis at the individual level identified a strong association between height and increased risk of atrial fibrillation, and decreased risk of coronary artery disease. MR analysis at both the population and individual levels suggested that height remains a causal risk factor for atrial fibrillation even after adjustment for other traditional risk factors.

Our findings are consistent with prior observational analyses that have identified height as a risk factor for atrial fibrillation [4,11–18]. These studies, including a large Swedish national cohort study of 1.5 million military conscripts recruited over the course of 28 years, have consistently identified a strong association between height and atrial fibrillation [4]. Observational designs have limited these studies, due to the possibility of residual confounding. The Helsinki Birth Cohort Study partially addressed this limitation by considering the effect of maternal height on risk of atrial fibrillation in offspring, but was limited by a small, homogenous sample [35]. We further build on those prior findings using the MR framework, considering both summary-level and individual-level genetic data, examining genetic variants associated with height and atrial fibrillation to provide a causal estimate for the effect of increasing height on risk of atrial fibrillation. Our population-level MR analysis reinforces a prior MR estimate for association between increasing height and atrial fibrillation (OR 1.33; 95% CI 1.26 to 1.40) using updated GWAS summary statistics for height and atrial fibrillation [9]. By including a large multiethnic study of atrial fibrillation, we increase the generalizability of the prior MR findings, which were limited only to white participants of the UK Biobank.

Several mechanisms have been proposed to explain the relationship between height and atrial fibrillation. Increased left atrial size is correlated with height, and has been identified as an independent predictor of atrial fibrillation [17,18,36,37]. Consistent with findings from the Cardiovascular Health Study, however, we found that the effect of height on risk of atrial fibrillation was not attenuated after adjustment for left atrial diameter [14]. It is possible that other, more nuanced markers of left atrial structure and function that have been associated with severity of atrial fibrillation, such as left atrial volume index, emptying fraction, expansion index, and contractile function, may also be affected by height and may better explain the association between height and atrial fibrillation [38]. Similarly, 2-dimensional echocardiography significantly underestimates left atrial size compared to MRI assessment [39]. Investigation of these factors in the 2-sample setting is limited by the lack of genetic studies of cardiac structure/function by echocardiography, MRI, or cardiac CT, and study of these parameters in large cohorts with genetics and imaging data is warranted. Bioimpedance and dual-energy X-ray absorptiometry measures of body composition have also been associated with atrial fibrillation [40]. While the current study focused primarily on common cardiometabolic risk factors used in clinical practice, the possibility remains that more advanced anthropometric screening beyond height, weight, body mass index, and waist-to-hip ratio may explain some of the effect of increased height on atrial fibrillation.

MR has previously identified an association between shorter stature and increased risk of coronary artery disease, mediated in part by increases in LDL cholesterol and triglyceride levels [10]. Our PheWAS of measured height similarly identified an association between height and decreased risk of coronary artery disease, and our PheWAS of a height GRS identified a nominally ($p < 0.05$) protective association between height and coronary artery disease. In the current MR analysis, we were unable to detect significant associations between these lipid traits and risk of atrial fibrillation when considered in multivariable models alongside height. Similarly, in multivariable models considering height alongside common cardiometabolic risk factors for atrial fibrillation, effect estimates were all similar, suggesting these factors may not substantially mediate the effect of height on risk of atrial fibrillation.

Although we detected no evidence of horizontal pleiotropy and our results remained robust to extensive sensitivity analyses, height is a highly polygenic trait, and the possibility remains that pleiotropic pathways mediate the association between height and atrial fibrillation. A recent GWAS found genes at atrial-fibrillation-associated loci to be enriched in pathways important for tissue formation [41]. Coupled with the finding that genes at height-associated loci are enriched for cardiovascular and endocrine tissue types, these results raise the possibility of a more complex shared genetic architecture affecting both height and atrial fibrillation [22]. Thus, we cannot exclude the possibility that genetic variants broadly associated with growth and development may simultaneously affect height and establish structural cardiovascular changes that may predispose to atrial fibrillation.

The findings of this study have several clinical implications. Anthropometric characteristics have been included in clinical models that predict incident atrial fibrillation, including BMI in a model derived from the Framingham Heart Study, and height in the CHARGE-AF (Cohorts for Heart and Aging Research in Genomic Epidemiology–Atrial Fibrillation) model from the CHARGE-AF Consortium [42–44]. While height is not a readily modifiable risk factor, the recognition that taller individuals have increased risk of atrial fibrillation may prompt more aggressive management of modifiable cardiovascular risk factors like overweight/obesity, pre-diabetes/diabetes, hypertension, and alcohol/tobacco use. As both height and atrial fibrillation are heritable, it is possible that height might act as a surrogate for family history of atrial fibrillation when this information is not readily available. Further study may clarify the risks of incident atrial fibrillation attributable to height and family history. Similarly, further mechanistic investigation of the height–atrial fibrillation link may identify novel modifiable pathways for intervention. Future study is needed to determine whether risk prediction tools including height or other anthropometric factors can be used to improve screening and primary prevention of atrial fibrillation.

Our study has several limitations. First, observational analyses from the Penn Medicine Biobank may not be generalizable, as this population represents a cohort within a single academic health system. Second, despite our use of multiethnic GWASs of height and atrial fibrillation, the underlying studies focused primarily on individuals of European ancestry. Genetic studies in broader populations are warranted to further improve the generalizability of these findings. Third, our population-level MR analyses relied on publicly available summary statistics that contain some overlapping samples/cohorts. Two-sample MR tends to bias the causal effect estimates to the null, but sample overlap may make the estimate susceptible to weak instrument bias. In this study, however, despite overlapping cohorts, simulation suggests that the large sample sizes of the height and atrial fibrillation GWASs, and large $F$-statistics, make the risk of weak instrument bias low [30].

In conclusion, we find that increased height is associated with increased risk of atrial fibrillation, and this relationship is likely to be causal. These results raise the possibility of investigating height/growth-related pathways as a means for gaining novel mechanistic insights into atrial fibrillation, as well as the possibility of incorporating height into larger targeted screening strategies for atrial fibrillation.

## Supporting information

**S1 Fig. Two-sample MR sensitivity analysis.** Two-sample MR was performed using a genetic instrument containing 224 independent SNPs associated with height, excluding SNPs nominally associated ($p < 0.05$) with traditional atrial fibrillation risk factors: coronary artery disease, HDL, LDL, total cholesterol, triglycerides, fasting glucose, fasting insulin, diabetes, BMI, waist-to-hip ratio, and systolic blood pressure. (A) Each point represents the SNP effects on

height and atrial fibrillation. Colored lines represent inverse-variance-weighted (red), weighted median (green), and MR-Egger (blue) estimates of the association between a 1-SD increase in height and risk of atrial fibrillation. (B) Odds ratios (ORs), 95% confidence intervals (CIs), and *p*-values for MR estimates.
(TIF)

**S2 Fig. Two-sample MR directionality sensitivity analysis.** Two-sample MR was performed using a genetic instrument containing 73 independent SNPs associated with atrial fibrillation. (A) Each point represents the SNP effects on atrial fibrillation and height. Colored lines represent inverse-variance-weighted (red), weighted median (green), and MR-Egger (blue) estimates of the association between a 1-SD increase in height and risk of atrial fibrillation. (B) Odds ratios (ORs), 95% confidence intervals (CIs), and *p*-values for MR estimates.
(TIF)

**S3 Fig. Individual MR sensitivity analysis.** Individual-level instrumental variable analysis was performed in the subset of Penn Medicine Biobank participants with clinically obtained echocardiogram data, using a GRS for height as an instrumental variable for measured height. The base model was adjusted for age, sex, and 6 genetic principal components. Model 1 was additionally adjusted for weight, hypertension, coronary artery disease, heart failure, hyperlipidemia, diabetes, chronic kidney disease, sleep apnea, stroke, thyroid disease, smoking, cardiac surgery, and valvular heart disease. Model 2 was additionally adjusted for left atrial size as measured on transthoracic echocardiogram. Odds ratios (ORs) are reported per 1-SD increase in height.
(TIF)

**S4 Fig. Genetic principal component scree plot for Penn Medicine Biobank.** Proportion of variance explained for each genetic principal component among European-ancestry participants of Penn Medicine Biobank.
(TIF)

**S1 Methods. Supplemental methods.**
(DOCX)

**S1 STROBE Checklist. STROBE checklist.**
(DOCX)

**S1 Table. Full genetic instrument for height used in 2-sample MR analysis including independent (distance threshold = 10,000 kb, $r^2$ = 0.001), genome-wide significant ($p < 5 \times 10^{-8}$) variants.**
(XLSX)

**S2 Table. Restrictive genetic instrument for height used in 2-sample MR analysis including independent (distance threshold = 10,000 kb, $r^2$ = 0.001), genome-wide significant ($p < 5 \times 10^{-8}$) variants, excluding those SNPs nominally ($p < 0.05$) associated with coronary artery disease, HDL, LDL, total cholesterol, triglycerides, fasting glucose, fasting insulin, diabetes, BMI, waist-to-hip ratio, and systolic blood pressure in the MR-Base database.**
(XLSX)

**S3 Table. Full genetic instrument for height used in individual-level MR analysis including independent (distance threshold = 10,000 kb, $r^2$ = 0.001), genome-wide significant ($p < 5 \times 10^{-8}$) variants.**
(XLSX)

**S4 Table. Results of bivariate multivariable MR analysis considering the effect of height and each of coronary artery disease, HDL, LDL, total cholesterol, triglycerides, fasting glucose, fasting insulin, diabetes, BMI, waist-to-hip ratio, and systolic blood pressure on atrial fibrillation.**
(XLSX)

**S5 Table. Results of combined multivariable MR analysis considering the effects of height, body mass index, total cholesterol, systolic blood pressure, and coronary heart disease.**
(XLSX)

**S6 Table. Summary demographics for Penn Medicine Biobank participants overall.**
(XLSX)

**S7 Table. Summary demographics for Penn Medicine Biobank participants stratified by quartile of height GRS.**
(XLSX)

**S8 Table. Summary demographics for Penn Medicine Biobank participants limited to subset of individuals with complete phenotype and echocardiographic data.**
(XLSX)

## Acknowledgments

The authors would like to thank the participants of the Penn Medicine Biobank.

**Regeneron Genetics Center banner author list**

All authors/contributors are listed in alphabetical order.

**Management and leadership team.** Goncalo Abecasis, PhD; Aris Baras, MD; Michael Cantor, MD; Giovanni Coppola, MD; Aris Economides, PhD; Luca A. Lotta, MD, PhD; John D. Overton, PhD; Jeffrey G. Reid, PhD; Alan Shuldiner, MD.

**Sequencing and lab operations.** Christina Beechert; Caitlin Forsythe, MS; Erin D. Fuller; Zhenhua Gu, MS; Michael Lattari; Alexander Lopez, MS; John D. Overton, PhD; Thomas D. Schleicher, MS; Maria Sotiropoulos Padilla, MS; Karina Toledo; Louis Widom; Sarah E. Wolf, MS; Manasi Pradhan, MS; Kia Manoochehri; Ricardo H. Ulloa.

**Genome informatics.** Xiaodong Bai, PhD; Suganthi Balasubramanian, PhD; Leland Barnard, PhD; Andrew Blumenfeld; Gisu Eom; Lukas Habegger, PhD; Alicia Hawes, BS; Shareef Khalid; Jeffrey G. Reid, PhD; Evan K. Maxwell, PhD; William Salerno, PhD; Jeffrey C. Staples, PhD.

**Research program management.** Marcus B. Jones, PhD; Lyndon J. Mitnaul, PhD.

## Author Contributions

**Conceptualization:** Michael G. Levin, Matthew C. Hyman, Saman Nazarian, Daniel J. Rader, Benjamin F. Voight, Scott M. Damrauer.

**Data curation:** Renae Judy, Shefali S. Verma, Yuki Bradford, Marylyn D. Ritchie.

**Formal analysis:** Michael G. Levin.

**Investigation:** Michael G. Levin, Matthew C. Hyman, Saman Nazarian, Benjamin F. Voight, Scott M. Damrauer.

**Methodology:** Michael G. Levin, Dipender Gill, Marijana Vujkovic, Benjamin F. Voight, Scott M. Damrauer.

**Project administration:** Renae Judy, Scott M. Damrauer.

**Resources:** Michael G. Levin, Renae Judy, Daniel J. Rader, Benjamin F. Voight.

**Supervision:** Benjamin F. Voight, Scott M. Damrauer.

**Writing – original draft:** Michael G. Levin.

**Writing – review & editing:** Michael G. Levin, Renae Judy, Dipender Gill, Marijana Vujkovic, Shefali S. Verma, Yuki Bradford, Marylyn D. Ritchie, Matthew C. Hyman, Saman Nazarian, Daniel J. Rader, Benjamin F. Voight, Scott M. Damrauer.

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
