## [Decision Letter · Decision Letter 0]

13 Feb 2020

Dear Dr. Damrauer,

Thank you very much for submitting your manuscript "Genetics of Height and Risk of Atrial Fibrillation: A Mendelian Randomization Study" (PMEDICINE-D-20-00041) for consideration at PLOS Medicine. 

[LINK]

In light of these reviews, I am afraid that we will not be able to accept the manuscript for publication in the journal in its current form, but we would like to consider a revised version that addresses the reviewers' and editors' comments. Obviously we cannot make any decision about publication until we have seen the revised manuscript and your response, and we plan to seek re-review by one or more of the reviewers. 

We expect to receive your revised manuscript by Feb 27 2020 11:59PM. Please email us (plosmedicine@plos.org) if you have any questions or concerns.

We look forward to receiving your revised manuscript. 

Sincerely,

Adya Misra, PhD

Senior Editor 

PLOS Medicine

plosmedicine.org

Article meta-data- some of the meta-data are incomplete and we request your attention to providing this information for instance- funding information, ethics statement, competing interests etc. Please contact plosmedicine@plos.org if you need assistance 

Abstract- Please structure your abstract using the PLOS Medicine headings (Background, Methods and Findings, Conclusions).

Abstract Background: Provide the context of why the study is important. The final sentence should clearly state the study question.

Abstract Methods and Findings:

* Please ensure that all numbers presented in the abstract are present and identical to numbers presented in the main manuscript text.

* Please include the study design, population and setting, number of participants, years during which the study took place, length of follow up, and main outcome measures.

* Please quantify the main results (with 95% CIs and p values).

* Please include the important dependent variables that are adjusted for in the analyses.

* Please include the actual amounts and/or absolute risk(s) of relevant outcomes (including NNT or NNH where appropriate), not just relative risks or correlation coefficients. (example for absolute risks: PMID: 28399126). 

Abstract-“as well as incorporating height into population screening strategies for atrial fibrillation” I’m not sure how screening strategies fit into your study so I would remove this from the conclusions 

Author summary : At this stage, we ask that you include a short, non-technical Author Summary of your research to make findings accessible to a wide audience that includes both scientists and non-scientists. The Author Summary should immediately follow the Abstract in your revised manuscript. This text is subject to editorial change and should be distinct from the scientific abstract. Please see our author guidelines for more information: https://journals.plos.org/plosmedicine/s/revising-your-manuscript#loc-author-summary

References- please provide a full stop after the brackets, for example [1]. 

Please provide 95% CIs and p values throughout and use standard notation such as p>0.001 or p=xxx if over 0.001 

The Data Availability Statement (DAS) requires revision. For each data source used in your study: 

Please ensure that the study is reported according to the STROBE guideline, and include the completed STROBE checklist as Supporting Information. When completing the checklist, please use section and paragraph numbers, rather than page numbers. Please add the following statement, or similar, to the Methods: "This study is reported as per the Strengthening the Reporting of Observational Studies in Epidemiology STROBE guideline (S1 Checklist)."

Comments from the reviewers:

Reviewer #1: In the manuscript entitled " Genetics of Height and Risk of Atrial Fibrillation: A Mendelian Randomization Study " the authors investigated the association of height with incidence of atrial fibrillation. 

They performed a two-step approach with an initial analysis using summary-level data for GWAS of height and atrial fibrillation. In a second step they verified the results using individual-participant data from the Penn Medicine Biobank and found similar results. Even after adjusting for relevant clinical factors the results remained consistent. The authors conclude that height is causally associated with high risk of atrial fibrillation. The study size is adequate and the statistics used are appropriate. The manuscript is well written and the topic is of scientific relevance and interest. However, I have a few major concerns and comments:

1. One concern is the missing link to the clinical relevance of the findings. How can these findings add to diagnosis, prevention or therapies in patients with AF in a clinically routine? e.g. a risk score? Should tall people try to lose weight? Can we try to compensate our stature? I think the manuscript would benefit from a section on this.

2. The finding that height and atrial fibrillation are causally related is interesting. Similar, fat-free mass has been described as an independent risk factor for atrial fibrillation (Tikkanen et al EHJ 2019). Regarding height and traits like fat-free mass, is there a genetic overlap? Do some of these variants affect both - height and fat-free mass? 

3. Did the authors see any gender differences between men and women in their analyses?

4. If data on LA-size is available, why was this not included in the baseline table?

Reviewer #2: General comments

Interesting and well conducted study that investigates the evidence for causality between height and incident atrial fibrillation. The authors state that their findings raises the possibility of investigating pathways of height. 

The authors also state that their findings suggest to use height in screening strategies for atrial fibrillation. That is not part of the research described in this paper, and seems overinterpretation of their results. 

Specific comments

Introduction

The background and aim of the study are discussed. Clear overview of the manuscript is given.

Materials and methods

Methods are discussed in a well-arranged manner. According to the supplementary data atrial fibrillation is identified by ICD codes in the Penn dataset. However, atrial fibrillation ascertainment should be added in the materials and methods section of the paper itself.

To accommodate the readers, please align the order of analyses in the methods and results section. In the results section the text starts with data of the individual-level analyses while the methods section starts with the summary-level two-sample MR. The same applies to the discussion section where the analyses on population-level are discussed first; try to arrange a certain order to describe the results.

Results

In most literature about height 'centimeter' instead of 'inches' is used. If the authors chose to use centimeters instead of inches, the data is easier comparable with existing literature. It may be worth using the SI system or both to accommodate worldwide readers. 

In the individual-level Mendelian randomization adjustment for 5 genetic principal components was choosen. Did the authors make a screen plot or was it an arbitrary estimate? Please provide a screen plot in supplementary data if possible.

Discussion

The authors also state that their findings suggest to use height in screening strategies for atrial fibrillation. That is not part of the research described in this paper, and seems overinterpretation of their results. Please remove.

Reviewer #3: The main claim of the paper is that developing atrial fibrillation is causally related to height. The main result is from a two-sample mendelian randomisation model, with data from two large GWAS studies whose summary data is publicly available. A clear scientific background is given, along with a rationale for investigating the association between height and atrial fibrillation. Due to the implausibility of conducting a randomised controlled trial with height as an exposure, Mendelian Randomisation is a sensible approach to take to investigate this association. 

This claim is not novel, there was a large study that demonstrated a very similar relationship (Lai FY, Nath M, Hamby SE, et al. Adult height and risk of 50 diseases: a combined epidemiological and genetic analysis. BMC Med 2018;16:187. doi:10.1186/s12916-018-1175-7). The claims made by that paper are properly placed in the context of previous literature. That paper looked only at people of European ancestry and did not do the level of sensitivity analysis shown here. This paper gives strong evidence to support this earlier paper, while also showing some generalisability to non-European populations. It also uses a secondary analysis on individual level data from Penn Medicine Biobank to support the main result and investigate some potential mechanisms. 

The prior study also found a multitude of other diseases associated with genetically determined height such as reduced CAD and increased hip fracture. This paper is interesting in that it didn't find any significant confirmation of those other results as part of its PheWAS, potentially due to a higher standard of correction for multiple testing being applied. Sadly, the authors do not go into much detail about which of these earlier results appear on the PheWAS at a lower threshold of significance or discuss whether they have any evidence to contradict the results of the earlier paper in these diseases. 

The main result supports the causal claim in the abstract, and the number and consistency of the sensitivity analyses give reassurance as to the robustness of these results. I have some concerns about some of the secondary analyses. I detail these below. The selection of the genetic variables is sensible and well-described; both in the primary analysis and the sensitivity analysis with a reduced genetic instrument. The choice of analyses to perform is comprehensive and well-chosen, and for the most part, reasonably well explained. The discussion of the limitations of this study was appropriate. The results are mostly correctly interpreted and compared sensibly with other relevant studies. The conclusion of this study was appropriate. The potential for clinical relevance is limited, due to the impracticalities of intervening on height, but the potential for investigating novel pathways or screening strategies is present. 

This study follows most of the relevant guidelines in MR-STROBE; there are some specific and easy to fix queries detailed below. The paper is reasonably well-organised and should be accessible to non-specialists.

The underlying research question is an important one, but given the earlier paper, it is not clear that these results are a substantial advance on existing knowledge. The results will be of interest to policymakers and clinicians, particularly those with an interest in screening strategies for atrial fibrillation.

This paper is well done, but I think it is held back from being outstanding by the combination of its overall similarity to an earlier study and the lower statistical rigour in the more novel parts. It would be improved by a higher focus on the individual level analysis and the potential mechanisms for the effect.

~~~

Major Concerns

I am not convinced by the statistical rigour of the secondary analysis on the individual level data. This is a shame, as the inclusion of this data and the opportunity it affords for investigation of the potential mediators and mechanisms of the effect under investigation is one of the more novel parts of this paper. 

The model is not sufficiently explained; I am somewhat confused as to what exactly has been done and what conclusions have been drawn from it. It would be helpful for the authors to expand this section, to clarify the hypothesis being tested and how they are interpreting the resulting odds ratios. 

[p9] "In the first stage of the two-stage process, a linear regression was conducted with standardized height as the dependent variable, and the

standardized genetic risk score for height as the independent variable. In the second stage, a logistic regression model with robust standard errors was fit, incorporating both the residuals from the first stage and scaled height, with atrial fibrillation as the outcome."

It is not clear whether the first stage was assessed only in controls, or if the cases were included as well. Including cases in this estimate could introduce bias to the model. I am unsure what scaled height refers to in the second stage but using an adjusted two-stage approach makes the odd ratio is very complex to interpret. https://www.ncbi.nlm.nih.gov/pmc/articles/PMC5642006/ does not recommend adjustment on first-stage residuals, and it would be useful to the reader to have greater clarity on how conditioning the IV estimate on the residual should be interpreted as a result. 

[p9] "In additional models, the second stage was adjusted for clinical diagnoses and left atrial size." 

This implies that the first stage was not adjusted for these covariates. In addition to the concerns above, this raises some questions over what the ORs in these additional models are representing when the second stage, but not the first is adjusted for covariates. Could the authors please clarify what hypotheses are being tested here? A better alternative would be to use the same covariates in the first- and second-stage IV regressions. If a specific conditional estimate is intended, then it needs much more explanation to the reader. See http://www.phpc.cam.ac.uk/ceu/files/2012/11/letterthirdone010611-1.pdf & https://www.bmj.com/content/362/bmj.k601 for further detail on this issue. 

It is also worth noting that the samples used for this MR analysis were also used in determining the instrument (as part of the "Multi-ethnic genome-wide association study for atrial fibrillation")

[P15] "Using individual-level MR analysis, we found that the relationship between height and atrial fibrillation appears to be independent of traditional clinical and echocardiographic risk factors for atrial fibrillation."

This is too strong a statement. I am unconvinced that this claim can be justified from the paper as written. Either the authors need to expand and explain the results that have led to this, or they need to soften this conclusion. 

Minor Concerns

MR Assumptions:

There is no discussion of the underlying MR assumptions & how they are justified. While some tests of the assumptions have been performed (e.g. MR-Egger to check for heterogeneity), these tests aren't explicitly linked to the underlying MR assumptions. It is critical for these assumptions & the evidence for them to be explicit so that the reader can use them to evaluate the validity of the model.

Information on the data sources for the two-sample MR:

The first line of the methods section says "Summary-level data for GWAS of height and atrial fibrillation were obtained." but it requires some effort from the reader to discover exactly where this data comes from. Based on the references in the previous section, the AF associations comes from the Nature paper "Multi-ethnic genome-wide association study for atrial fibrillation", which has data from UK Biobank, Biobank Japan, & others - including Penn Medicine Biobank. The height associations come from "Meta-analysis of genome-wide association studies for height and body mass index in ∼700000 individuals of European ancestry", which shows that the data for height comes from a meta-analysis of data from UK Biobank & Wood et. al (which takes data from 79 smaller studies).

Firstly, the reader should not have to read 3-4 references to determine the underlying population of the study. Please provide a clear account of the source of the data, it's underlying population etc. This is particularly relevant as the multi-ethnicity of the populations is one of the strengths of the study. Please include results on the heterogeneity of the different populations of your various data sources - the table for the Penn Medicine Biobank is excellent and it would be helpful to have as similar as possible for the populations that the AF & height data came from. A table in on the individual level data comparing the 25th and 75th percentile of your genetic risk score for the Penn Medicine Biobank data could also provide useful context for considering the multivariate genetic effects.

Also, UK Biobank's European participants are included in both studies, giving an overlap of at least 50%. This is briefly mentioned in the limitations section of the discussion, but a clear statement as to the level of the overlap & its estimate on the impact of this overlap should be given. https://sb452.shinyapps.io/overlap/ may be useful here. 

There is no comment on how the summary statistics were calculated for each data source - were similar covariates included in both calculations?

Multiple definitions of AF

Different definitions of AF are used by different data sources. e.g. from "Multi-ethnic genome-wide study...": "Ascertainment of AF in the UKBB includes samples with one or more of the following codes: non-cancer illness code, self-reported (1471, 1483); operation code (1524); diagnoses - main/secondary ICD10 (I48, I48.0-4, I48.9); underlying (primary/secondary) cause of death: ICD10 (I48, I48.0-4, I48.9); diagnoses - main/secondary ICD9 (4273); operative procedures - main/secondary OPCS (K57.1, K62.1-4)" whereas the definition used in the Penn Medicine Biobank definition is "Atrial fibrillation was defined using ICD9/10 codes: 427.31, I48.0, I48.1, I48.2, I48.91.". I do not think this is problematic, but it should be made clear to the reader.

Typos/Corrections

In methods, it says "The instrumental variable was a standardized genetic risk score for height, computed from

independent, genome-wide significant variants, weighted by the effect on height in the GIANT

GWAS (see Height Genetic Risk Score above)." (P9) but the section on Height Genetic Risk does not appear until P11. 

Multiple testing was managed via Bonferroni correction in the PheWAS analysis, but this is only stated on the relevant figure. Please make it clear in the body of the text also.

Figure 3 is a little unclear. I think it is plotting the OR of Height on AF but there is no scale provided (presumably per 1S.D. change in height?) from the Multivariable MR that also includes the variables named under covariate. It feels odd to describe the other risk factors includes in the multivariable MR as covariates - that implies more a measured confounder to my ears. The OR for the other risk factors should also be reported (some are in ST4, but not for the combined model) 

p13 says "Inverse variance weighted modelling identified a significant association between increasing height and atrial fibrillation (OR 1.34; 95% CI 1.29 to 1.40; p = 5x10-42)" could the authors first clarify what the scale of height is for this OR - I am presuming per S.D. increase as with the Phenome-wide association studies, but it would be useful to have it explicitly. Could the authors also please translate this into a clinically understandable result - e.g. what is the increased risk for a patient who is 10cm taller.

ST2-4: please include s.e. and p-value for each odd ratio, not just the 95% confidence interval.

[LINK]

---

## [Decision Letter · Decision Letter 1]

7 Jul 2020

Dear Dr. Damrauer,

Thank you very much for re-submitting your manuscript "Genetics of Height and Risk of Atrial Fibrillation: A Mendelian Randomization Study" (PMEDICINE-D-20-00041R1) for review by PLOS Medicine.

I have discussed the paper with my colleagues and the academic editor and it was also seen again by two of the original reviewers. I am pleased to say that provided the remaining editorial and production issues are dealt with we are planning to accept the paper for publication in the journal.

[LINK]

We look forward to receiving the revised manuscript by Jul 14 2020 11:59PM. 

Sincerely,

Thomas McBride, PhD

Senior Editor 

PLOS Medicine

plosmedicine.org

Requests from Editors:

1- Please consider different wording than “leverage(d)” in the Abstract and elsewhere (lines 42, 70, 105, 229).

2- In the Abstract, please provide the years of enrollment for all of the cohorts, as well as additional demographic information (e.g., sex). 

3- Thank you for including the study limitations in the Abstract. Please remove the word “may”. Also, “*lack of* generalizability”?

4- Please expand the Abstract Conclusions to include the potential implications. While true that these findings do not provide evidence for incorporating height into screening strategies, you could note that that they suggest future studies should investigate “whether risk-prediction tools including height or other anthropometric factors can be used to improve screening and primary prevention of atrial fibrillation”, as noted in the Discussion.

5- Please also edit the Abstract Conclusions to limit the conclusions to the current study. "In this study, we observed ..." may be useful.

6- Thank you for adding an Author Summary. On line 66, please replaces “actually causes” with “elevates the risk of”

7- On line 73, please edit to: “Genetic variants associated with taller stature *were also associated with* increased risk of atrial fibrillation.”

8- Thank you for providing your STROBE statement. Please replace the page numbers with paragraph numbers per section (e.g. "Methods, paragraph 1"), since the page numbers of the final published paper may be different from the page numbers in the current manuscript.

9- Is it possible to include the graph from your response to reviewer 2, comment 6 as a supplemental figure?

10- Did your study have a prospective protocol or analysis plan? Please state this (either way) early in the Methods section.

11- Please remove the “Transparency” section.

12- Please move the Ethical approval statement to the Methods section. Was approval sought or required for the specific analyses in this study? If so please specify in this section. Additionally, it seems odd that the “biobank” received approval, do you mean the researchers at the biobank, including the authors of this study?

13- Please check the formatting for reference 3 

14- References 19 and 23 seem like different versions the same paper, one of which is a preprint. Please reconcile.

Comments from Reviewers:

Reviewer #1: The authors have addressed all of my concerns satisfactorily with great benefit for the manuscript. I am happy to recommend the study for publication in Plos Medicine. 

Reviewer #4: As the fourth reviewer, I have seen the revised version of the manuscript and only have a few minor comments which appears to much more detailed and improved than the original. 

The methods are very complete, designed to test to the robustness of the authors assumptions and results. The separate components of the study (MR two stage, individual-level MR, and PheWAS) all confirmed consistency in results. From this analysis the conclusion that height is causally associated with AF is confirmed. 

Minor comments:

Lines 131 - 133: Can the authors describe what method they used for allele harmonisation to ensure exposure and the effect of that SNP on the outcome corresponds to the same allele

Lines 141 - 144: Whats the rationale for excluding SNPS using a nominal p-value of 0.05? 

Lines 186-187: was 10 PCs determined a priori or through investigation of the variation of the data. 

Line 211: How were potential confounders/mediators selected?

Lines 419-421: In terms of clinical implications for risk prediction models for AF - many do already include anthropometric factors, including the recent study in JACC (https://pubmed.ncbi.nlm.nih.gov/31706453/) which includes both height and weight. I guess one could say that the results confirm that this is the correct choice to include as a potential a priori factor - but how much height contributes to discrimination of the outcome is likely to be pretty modest. I think the key clinical implication still remains that this study opens the door for mechanistic work to understanding the biological pathways which more has implications in furthering therapeutics development

Final comment: What do the authors think about family history of AF - if height is causally associated with AF, and height has inheritability, it would make sense that a taller person may have more likelihood of family history of AF. Any implications for the study results in terms simply assessing family history for AF.

[LINK]

---

## [Editor Report · Decision Letter 2]

3 Sep 2020

Dear Dr. Damrauer, 

On behalf of my colleagues and the academic editor, Dr. Michiel Rienstra, I am delighted to inform you that your manuscript entitled "Genetics of Height and Risk of Atrial Fibrillation: A Mendelian Randomization Study" (PMEDICINE-D-20-00041R2) has been accepted for publication in PLOS Medicine. 

PRODUCTION PROCESS

PRESS

PROFILE INFORMATION

Thank you again for submitting the manuscript to PLOS Medicine. We look forward to publishing it. 

Best wishes, 

Thomas McBride, PhD

Senior Editor 

PLOS Medicine

plosmedicine.org